# Perception of Spatial Legibility and Its Association with Human Mobility Patterns: An Empirical Assessment of the Historical Districts in Rasht, Iran

**DOI:** 10.3390/ijerph192215258

**Published:** 2022-11-18

**Authors:** Reza Askarizad, Jinliao He

**Affiliations:** The Center for Modern Chinese City Studies, Institute of Urban Development, East China Normal University, Shanghai 200062, China

**Keywords:** cognitive sketch maps, human mobility, spatial legibility, Space Syntax, urban image

## Abstract

Achieving legibility within the context of historical districts has become a controversial problem due to their widespread growth and unconventional constructions within, which has led to inconsistencies in the urban context system, and a decrease in the level of urban sociability. This paper aims to provide an empirical assessment towards facilitating the perception of spatial legibility and its association with human mobility patterns. To this end, a novel mix method was developed in order to comprehend the association between spatial legibility and human mobility patterns using Space Syntax, cognitive sketch maps, and time-lapse photography. The results revealed that there is a significant association between spatial legibility and human mobility patterns, such that the incorporation of objective and subjective factors affecting legibility, including highly integrated morphological characteristics along with the saliency of landmarks featuring historical values, can lead to increased human mobility patterns in terms of use frequency. Accordingly, this research aids urban planners and designers in recognizing how to deal with historical districts in order to foster the sociability of these areas and create a lively and socially sustainable urban environment.

## 1. Introduction

Legibility is one of the most important features of high-quality urban spaces, and it can be defined as the ability to organize the environment using coherent and imaginative patterns [1]. Numerous studies have shown the importance of spatial configuration in spatial cognition [2]. The quality of legibility in an urban space can create a memorable journey for the audience, so that it facilitates spatial perception [3]. In this process, spatial intelligibility is the degree of our vision from the spaces of a complex, which is a qualified indication for the range of spatial perception and legibility [4]. Spatial intelligibility represents the amount of spatial information that can be visually obtained from an axial line [5]. In fact, studies have demonstrated that intelligibility can be considered as an indicator of legibility in urban areas [6]. In other words, the more intelligible an area, the higher the correlation between spatial configuration and movement patterns [2].

In the meantime, Space Syntax has been considered as one of the most outstanding research procedures in urban design, which incorporates a set of mathematical equations and organized propositions in order to assess social and movement behavioral patterns within the urban spatial configuration [7]. As a method, Space Syntax describes the topological relations of spatial configurations with respect to metric distances, analyzing them both mathematically and theoretically [8]. The mathematical parameters can be used to create a model that predicts the performance and behavior of people within an urban context [9]. The degree of intelligibility is effective in predicting spatial distribution in the minds of pedestrians [10]. In addition, studies have revealed that configurational facets of space have a significant correlation with cognitive mapping systems [4]. In fact, the Space Syntax theory focuses on the structure of the city, based on pedestrian vision in urban spaces [11]. In this process, components not only have pairwise relations with one another, but there is also a relationship among the overall structural patterns of the city [12].

On the other hand, cognitive maps can be considered as a sort of mental structure that comprises all the endogenous procedures that enable individuals to obtain information regarding the nature of their physical built environment [13]. These mental images are constantly being updated, so that they can act as a reference snapshot in the minds of individuals in order to evoke the physical characteristics of the built environment [14]. Therefore, it can be considered as a fundamental platform to serve as a guideline when a wide range of data is missed [15]. Cityscape Perception is defined as a prerequisite state of human experiences through the use of contemplation, imagination, and visualization; which can be disparate for different individuals [16]. The perceptual experience of cityscapes is scrutinized by expert-led landscape identity assessments through emphasizing all senses, encompassing sight, hearing, smell, touch, and taste, which can stimulate subjective influences [17,18,19,20,21].

Accordingly, arts-based qualitative research methods are considered as one of the most cutting-edge tools that can be adopted to obtain a set of rigorous data in respect to spatial cognition and perception [22,23]. This method provides opportunities to acquire a set of subjective data which are strictly immeasurable using surveys [24]. In this regard, sketch map drawings are deemed as a suitable method for evaluating individuals’ mentality in the field of cognitive psychology [25]. The perception of a mental map created in the minds of visitors is considered noteworthy when it comes to identifying desirable and undesirable landmarks [26]. Empirical studies have substantiated that every individual has a visual image of a city, and this image defines the city identity, structure, and meaning for that individual [27].

The study of historical districts and their values has always been one of the most significant discussions in architecture and urban development. Achieving legibility in contemporary cities has become a major issue. This is mainly due to the widespread growth of fragmented actions and unconventional constructions, which has led to inconsistencies in the urban context system. Unfortunately, in most urban planning projects, the concept of legibility is simplified down to a number of elementary solutions to facilitate orientation and wayfinding. The fact remains that this concept has many aspects and is quite comprehensive, and paying particular attention to them can lead to improving the richness of experience and perception of individuals from urban milieu. Hence, in this study, the concept of legibility has been considered in order to assess its association with the mobility patterns of people in the central part of Rasht City, which is located in the north of Iran. In this regard, attempts were made to provide an appropriate answer for the following questions:What is the role of spatial configuration on the legibility of the studied historical district?What is the role of historical districts in shaping cognitive maps in the minds of pedestrians?To what extent are the objective and subjective factors of legibility influential in shaping human mobility patterns?

The identification and perception of a spatial layout from the viewpoint of pedestrians has been extensively discussed by scholars so far [28,29,30,31,32,33,34,35,36,37,38]. Additionally, a wide range of studies have been conducted in order to evaluate the role of legibility in urban spaces using the Space Syntax method [6,39,40,41,42,43,44,45,46,47]. Moreover, a set of studies have been focused on the role of spatial cognition using sketch drawings [25,48,49,50,51,52,53,54]. In addition, in order to attain a suitable method towards assessing the movement patterns of people in urban spaces, a number of studies have culminated their works using time-lapse methods [55,56,57]. The present study seeks to bridge the identified scientific gap in order to identify a meaningful connection among spatial legibility, landmark, and movement patterns in urban spaces by adopting a mix method comprising Space Syntax, sketch maps, and time-lapse procedures. In this regard, the association among the aforementioned items would be designated as the main contribution of this study. This research contributes to urban planners and designers by helping them recognize how to deal with historical textures in order to foster the spatial legibility and sociability of these areas.

## 2. Research Design and Procedures

In order to provide an empirical assessment towards facilitating the perception of spatial legibility in the studied historical district, three major procedures have been adopted based on the given research questions. Accordingly, a mixed approach comprising both quantitative and qualitative methods has been implemented, namely Space Syntax, cognitive sketch maps, and time-lapse photography, in order to appraise the role of spatial configuration, landmarks, and human movement patterns in spatial legibility. The overall interrelation among the adopted procedures and research questions has been depicted accordingly (Figure 1). In the following subsections, fastidious descriptions are provided in respect to each adopted method.

### 2.1. Space Syntax Method and Its Role on Objective Spatial Legibility

The Space Syntax method, introduced by Bill Hillier et al. [58] at University College London, is founded on the notion that the physical characteristics of the built environment imply the way a space is explored, comprehended, and experienced from the viewpoint of pedestrians. Space Syntax assists urban designers and planners in understanding the role of spatial configuration in shaping people’s behavioral patterns, and predict the social function of the built environment by delving into a set of spatial attributes [59]. This procedure provides an opportunity to investigate the casual relationships between spatial configuration and its modalities of occupations [60]. Space Syntax literature has defined the concept of intelligibility as the capability of a space to give clues toward the perception of the urban spatial configuration objectively [6]. Studies have validated that there is a significant correlation between the degree of intelligibility and the value of spatial cognition [32,34,61,62].

In the context of Space Syntax literature, the values of integration and connectivity can be deemed as primary variables affecting a legible built environment. A higher value of integration is demonstrative of the frequent usage of that particular space by pedestrians, and a higher degree of connectivity can be an indicator of better spatial connection within that space [59,63]. Moreover, numerous studies have confirmed the fact that the correlation between integration and connectivity eventuates in spatial intelligibility [4,6,64,65,66,67,68]. Accordingly, the higher the value of the correlation, the higher the value of legibility is achieved. Hence, in the present study, the concepts of integration, connectivity, and intelligibility have been scrutinized using the Space Syntax method in order to assess the legibility of the historical texture in question based on its spatial configuration. To achieve this end, the Depthmap 10 software, developed by Alasdair Turner in London, UK [69], was used to launch the simulation analysis.

According to previously conducted studies [70], the best syntactical analyses for the applicability of legibility, and particularly the syntactical image of the city, include axial lines and Isovist. The axial line map performed as the most ideal descriptor, since it is the most common analytic procedure in the Space Syntax repertoire, and empirically the most validated one in terms of wayfinding in urban areas [5]. Nonetheless, Lynch’s accentuation on the visual attributes of urban elements provokes the possibility of using another alternative for spatial representations, and Isovist manifests as a potentially effective tool for this purpose. Isovist is defined as the field of view which is available from a particular vantage point, normally taken at eye height and parallel to ground level [71]. In fact, it facilitates the opportunity for seeing and being seen from a particular position.

The method of implementing the Space Syntax in this research is based on the initial scheme of urban development in the city of Rasht, which acquired its historical configuration during different dynasties. This procedure provides an opportunity to identify the effect of urban growth in different periods on spatial legibility. Since the primary focus of this study is on the historical districts of Rasht, attempts have been made to analyze urban developments in different periods. To achieve this aim, following the historic urban map of the city in the three dynasties of Safavid, Qajar, and Pahlavi, the relevant urban plans were drawn in AutoCAD 2010. Afterward, by importing the files into the Depthmap 10 software, the syntactical analysis in different periods was carried out in order to understand the impact of spatial configuration on spatial legibility. It should be noted that the process of calculation for the edge effects in this method has been clearly clarified in previously published sources [71,72].

### 2.2. Cognitive Sketch Maps Method and Its Role on Subjective Spatial Legibility

The theory of spatial legibility, and the assessment procedure of subjective mental imagery was initially introduced by Kevin Lynch [27]. His empirical studies, carried out in a number of American cities, were pioneering explorations to evaluate the cognitive map shaped in the minds of citizens using hand-drawn sketch maps. In this regard, Lynch [27] identified five main elements that affect the mental imagery of citizens within urban settings including paths, edges, districts, nodes, and landmarks. The hand-drawn sketches method can find its optimum reliability when it comes to the overlap among the outcomes of the acquired data [73]. Since all the sketch drawings followed a particular set of objectives, particular attention must be paid to the acquired differences in order to better infer the collected data [74].

From a methodological perspective, different approaches have been adopted to analyze the data collected through sketch drawings so far. Initially, scholars most often avoided using computational approaches, and preferred to manually analyze their data in favor of qualitative approaches [75]. Other studies employed independent processors in order to assign subjective ratings of the acquired sketch drawings for labeling landmarks, roads, and the other influential elements affecting urban imageability [76,77]. Moreover, cautiously designed rubrics have been considered in other studies [78,79]. More recent studies have concentrated their analysis using specialized computer software packages such as SketchMapia [80,81], Gardony Map Drawing Analyzer [82], Aram Mental Map Analyzer [83], and MaxQDA [50,84,85].

The present study carried out in-depth interviews with a community of experts in the fields of architecture, urban design, and graphic design using the cognitive sketch drawing method as the measuring instrument. Prior to conducting interviews, the authors ensured the participants’ drawing skills and capabilities by observing samples of their works. It must be declared that all the participants were natives and fully familiar with the socio-cultural characteristics of the studied historical texture. The method of data acquisition involved participants being initially asked to close their eyes and visualize the most prominent mental image that was shaped in their minds in respect to the cityscape of the studied area. Subsequently, the participants were asked to convert their subjectivity to objectivity through drawing their cognitive mental maps on sheets of paper in about 20 min in order to provide an opportunity for boosting their subconscious to develop their sketch drawings. This procedure was adopted in order to avoid any sort of bias or inaccurate orientation in the minds of the participants. The process of interviewing continued until appropriately saturated responses were received, and the process was halted once 33 participants had come up with their illustrations.

It should be noted that the MaxQDA10 software was used to qualitatively analyze the data obtained from the hand-drawn sketches. In this research, the cognitive sketch map analysis was categorized and coded based on Lynch’s mental imagery cityscape elements comprising *paths*, *edges*, *districts*, *nodes*, and *landmarks*. Afterward, the resembled codes were put together in their own classifications and compared with one another. Next, by quantifying the qualitative obtained codes, attempts have been made to analyze the final data in order to discern the most influential cityscape within the study area. It should be noted that this research does not merely aim to reproduce the conventional method that Kevin Lynch disseminated many years ago, but also intends to use the five main urban elements identified by him in order to codify and quantify the research data in a systematic manner. Some examples of the obtained cognitive sketch drawings are presented in Figure 2. Furthermore, the present research was conducted according to the rules of the *Declaration of Helsinki*. Since the present survey was anonymous, voluntary, and excluded any demographic data from the participants, the *Institutional Review Board* approval has been waived for this study.

### 2.3. Time-Lapse Method and Its Role on Human Mobility Patterns

The concept of human mobility patterns can be defined as the ability to move and interact freely and easily among people, which can be assessed through empirical analysis [86]. Time-lapse photography is a technique in which serial photos are captured at regular time intervals to determine the mobility patterns of what is being observed. Thus, it can be deemed as a kind of empirical observation method. This method is widely applied as an appropriate instrument in the field of social sciences in order to investigate cases in which the alteration in time matters. Studies have validated that the time-lapse technique is considered as an outstanding method for monitoring human mobility patterns [87]. This method also provides an opportunity to control a very long status over a period of time, and represent it in a much shorter time [88]. Additionally, this method offers precise and reliable data in order to ensure the validity of the obtained data over the time by considering the fourth dimension, which indicates its superiority compared to other conventional methods such as *gate count*. Accordingly, the time-lapse method is a privileged source of data which is easy to operate and provides stringent results in relation to human mobility patterns.

The method of implementing this procedure was based on the data obtained from the hand-drawn sketches; so that by identifying the most important landmark of the city, the position for capturing pictures has been determined in the study area. Accordingly, pictures were taken every 15 min between 5 pm and 10 pm in order to assess the human mobility patterns in front of the main landmark of the city. In this regard, the association between the main landmark of the city, as one of the most significant factors facilitating spatial legibility, and the movement patterns of people, have been evaluated. Moreover, the correlation between the time and movement patterns of pedestrians were investigated in order to discern whether there is a relationship between different time intervals and mobility patterns. It should be noted that the empirical experiments took place on an ordinary weekday (Saturday, 30 July 2022), in clear and sunny weather conditions, in order to eschew any intervening factors affecting observations. Moreover, all the recognizable faces have been blurred in order to ensure the anonymity of the participants.

### 2.4. Study Area

The city of Rasht, located in Gilan province, is one of the major metropolises in Iran, and is recognized as the third most visited city in the country. The historic value of this city dates back to the pre-Islamic period and the Sassanid era. The main core of urban development in this city was shaped in the Safavid, Qajar, and Pahlavi dynasties (Figure 3) [89]. The physical structure of the historical texture implied in this study focuses on the initial formation of this city, which still remains steady due to its unique and organic configuration. Thus, the central part of the city, which had been the initial core of urban development, is recognized as the studied historical texture in this research. The main reason behind the selection of the present case study is due to its historic values, and the sense of belonging and emotional attachment of citizens in this context; which, despite its old configuration, is still considered as the main promenade of the city, and a great deal of people visit these areas.

## 3. Results

### 3.1. Findings Obtained from Spatial Configuration Analysis

The main focus of the spatial configuration analysis was on the central core of the city, which encompasses the initial formation patterns of urban development, and embraces the historical contexts of the case study in different dynasties. The process of analysis commenced by scrutinizing the central part of the spatial configuration of the urban plan with a diameter of 3000 m. Afterward, by concentrating on the main central neighborhood, the analysis was performed covering a diameter of 1500 and 750 m, respectively, in order to compare the value of spatial legibility in different cases (Figure 4). The results indicated that the main axes leading to the central square of the city characterized the highest levels of integration values. This means that according to the morphological analysis, these streets possess the most capacity to attract visitors. This can be approved by the fact that the ultimate gathering destination for crowded communities is considered to be the main square of the city.

Further elaborated interpretations obtained from the syntactical analysis highlight the influential role of some specific paths. Initially, the highest integrated space, with the numerical value of 3.56, corresponded to the area in front of the *Municipality Building*, which is the most outstanding landmark of the city. Down the list are three main pedestrian paths leading to the *Central Plaza*. The first one, which is located in the northern part of the plaza, features the integration value of 3.25. The *Post Office* building, which is considered as a major landmark on this path, is located nearby. The second one, which is located in the western part of the plaza, has an integration value of 3.39. Likewise, with the quantitative value of 365,582, this path encompasses the highest range of choice, meaning it may be frequently used by pedestrians who are familiar with the study area. The *National Library* building is located adjacent to this path, and is considered as one of the milestone monuments associated with this road. The third path, which is located in the southern part of the square, features the integration value of 3.25. In addition, with an integration value of 3.13, the path leading to the *Great Bazaar* of the city is considered as another influential path in the study area that is frequently used by pedestrians. Hence, the identification of important paths which may be influential on the creation of cognitive maps in the minds of citizens were clarified.

Thus, the focal setting for the gathering of crowded communities is significantly associated with the location of the historical context within the entire urban spatial configuration. The results obtained from the correlation test between the integration and connectivity values in the diameter of 3000 m determined that the value of R^2^ was equal to 0.30, which indicates a very low status of spatial legibility based on the spatial configuration analysis. Moreover, the findings obtained from the correlation test indicated that the level of intelligibility in the diameter of 1500 m was equal to 0.42. In addition, the value of intelligibility in the diameter of 750 m revealed the proliferating range of spatial legibility by an R^2^ value of 0.60. The graphs of the adopted correlation tests between integration and connectivity have been presented in Figure 5. Observing the trend of intelligibility values in the different analyzed cases illustrated that the more focus there is on the local area, the higher the spatial legibility achieved.

According to the conducted analysis, it was intended to further investigate the central square in terms of its visual capabilities using Isovist graphs in order to understand which positions provide a better field of view. Since the notion of spatial legibility is strongly associated with the characteristics of seeing and being seen [27], Isovist graphs may offer the most potential field of view within the main milieu, which can lead to the creation of crowded communities in a legible environment. Amongst three positions that were spotted in the main square which possess a better vantage toward the urban landmark, the one in the middle offers the best field of view with the Isovist area of 26,853. Next, the spot of Isovist 3 was ranked second with the numerical value of 22,623. Thereafter, the spot of Isovist 2 was ranked third with its quantitative value of 20,712 (Figure 6). Hence, the best location for observing human mobility patterns, that can offer a better visual platform, was determined to be the middle part of the square.

### 3.2. Findings Obtained from the Hand-Drawn Sketches

The results indicated that the most frequent code derived from the hand-drawn sketches (a use of 31 times) was the *landmark*. Next, the *edge* (used 22 times), and the *path* (used 19 times) were considered as the most frequent factors affecting subjective legibility, respectively. In addition, the *node* and the *district* were considered as the least frequent signs in the cognitive maps of the participants. As is evident, the landmark was identified as the most important factor affecting the cognitive map of the participants (Figure 7). Thus, further analysis was conducted in order to delve into the identified landmark typologies in order to discern their particular attributes.

In relation to the prominent landmarks within the study area, three major criteria can be considered for identifying landmarks. The first is related to a set of administrative monuments, which have been constructed by the order of Reza Shah Pahlavi, and are considered as a milestone for the formation of the city’s historical context. The second criterion belongs to the vernacular elements and statues, which were set in place after a recent renovation project was accomplished in the study area. The third item can be referred to the historical bazaar of the city, which is considered as the cornerstone of economic prosperity in the study area.

Accordingly, the results obtained from analyzing the cognitive maps of the participants indicated that the *Municipality Building* was recognized as the most important landmark within the study area in terms of the frequency of utilized codes (having been depicted 12 times). Following this, the *vernacular elements and statues* (used 9 times), and the *post office building* (used 5 times) were placed in the second and third rank, respectively. It is worth mentioning that the *fountain* found in the center of the square, the *great Bazaar*, and the *National Library Building* were identified as the other landmarks identified from the cognitive maps of the participants, respectively (Figure 8).

### 3.3. Findings Obtained from Human Mobility Patterns

Based on the findings acquired from the cognitive sketch drawings, the *Municipality Building* has been identified as the most important landmark of the city. In addition, according to the conducted syntactical and Isovist analysis, the vicinity of this building was recognized as the most integrated space in terms of spatial configuration attributes. Therefore, this location was recognized as the most important position for implementing time-lapse photography in order to measure human mobility patterns in the course of time. Observations were conducted in the afternoon between 5.00 p.m. and 10.00 p.m., which are considered to be the most crowded hours in the study area. Photos were captured at a regular interval of every 15 min in order to follow up the possible transformations in the crowded communities forming in front of the main landmark of the city (Figure 9).

In the next step, by delving into the human mobility patterns observed in different time periods, attempts have been made to quantify the qualitative data obtained from the time-lapse photography method. According to the acquired results, the data illustrated the fact that as night approaches, the presence of people in this particular urban space increases. This fact can be substantiated by the observed cumulative trend line of the conducted chart in relation to the frequency of human mobility patterns (Figure 10). Moreover, the findings indicated that the least range of human mobility patterns were observed at 5.00 p.m., whereas the most crowded human mobility patterns were observed at 9.00 p.m., 8.15 p.m., and 9.15 p.m., respectively. It should be noted that the least crowded mobility pattern observed in front of the main landmark of the city included 65 people, and the most crowded mobility pattern observed at the same location included 220 people.

## 4. Discussion

### 4.1. Understanding the Contribution of the Present Study

The findings obtained from the spatial configuration analysis indicated that the local scale of legibility is considerably higher than its value in a larger scale within the study area. This issue elucidates the fact that the urban configuration network of the present case study is inconsistent with the intelligible attributes of an urban area, and the process of spatial cognition is constrained in local districts. This issue is in line with previous studies looking into the impact of underlying scaling of city artifacts on the creation of *the image of the city* [90]. As a result, there seems to be an association between the configurational aspects of legibility and the location of historical monuments in the study area. This finding is also aligned with previously conducted studies, which confirmed that an irregular spatial configuration may possess a lower value of legibility [91,92,93]. In addition, the results were found to be consistent with respect to prior studies, which affirmed that urban development can lead to decreasing levels of legibility and accessibility [44,94,95].

Consistent with previous studies [38,49,96,97,98], *landmark* was identified as the most important physical element affecting the subjective aspect of spatial legibility disseminated by Kevin Lynch [27]. In addition, three major criteria, including *historical monuments*, *vernacular elements*, and *socio-economic centers* such as traditional bazaars were found to possess an overwhelming impact on the creation of cognitive maps. Moreover, a number of previous studies found that the theory set forth by Lynch is limited in terms of the physical elements of the built environment, and ignored the spatial relationship between these elements [99,100], whereas this relationship is deemed to be fundamental for indicating the cognitive maps of people, and is considered as a precursor of the human mental image which can be appropriately addressed by the Space Syntax methodology [5].

Accordingly, several studies implemented a combination of the Space Syntax method and sketch map acquisition as their intended method for tackling this limitation, and confirmed that there is a significant association between configurational attributes and the saliency of landmarks based on the participants’ perspectives [6,32,42,101,102]. However, no studies have been carried out to evaluate the perception of spatial legibility and its association with human mobility patterns using a combination of syntactic, cognitive, and time-lapse photography methods. In this regard, in order to understand the relationship among the objective and subjective facets of spatial legibility and human mobility patterns, a set of empirical studies were accomplished as the most novel contribution of the study at hand to the existing literature in this field.

The most important reason for combining such procedures was to develop a framework which is able to ameliorate the limitations of previously conducted methods in order to identify the relationship between the perception of spatial legibility and human mobility patterns. Thus, this research has not sought to reproduce or imitate the adopted method that Kevin Lynch disseminated many years ago. On the contrary, it intended to take advantage of each adopted method to cover the shortcomings that alternative methods suffered from. For instance, the effect of spatial configuration on legibility, which was ignored in Lynch’s method, was covered by the Space Syntax method; or the limitation of focusing on 2D objective attributes in the syntactical analysis was compensated by the 3D subjective mental images of the participants. Likewise, the obtained mental images of the participants were converted into cognitive maps to reconcile with the urban layout. Moreover, the lack of empirical studies in the simulated graphs were tackled by temporal human mobility patterns to understand whether there is a relationship between legible areas and human agglomeration. Hence, this research seeks the extensions of procedures and instruments to ensure obtaining more thorough and meaningful datasets.

One of the advantages of applying the present research procedure compared to previous ones is that it elaborates empirical studies in different time intervals. It also provides a suitable platform for grasping the applicability of a livable urban environment at night time in comparison to daytime. Moreover, regardless of the qualitative nature of its research method, it offers an appropriate condition to convert the acquired data into quantitative ones. Thus, the present research method may suggest more precise results in respect to gauging human mobility patterns in a particular zone compared to conventional methods. These notions can be considered as the original contributions of the present research to the existing body of scientific literature.

### 4.2. Implications for Policy and Planning

In relation to the implications for policy and planning, there are some notions that need to be taken into consideration. The formation of historical districts mainly featured irregular spatial configurations, which in most cases led to the diminution of their accessibility and legibility. In this regard, most visitors, especially the ones who are unfamiliar with the intended destination, become profoundly confused during their wayfinding process. According to previously conducted studies, this issue can lead to increased stress and anxiety levels among the users of public spaces [103,104,105]. This problem is assumed to be tackled by measuring the intelligibility value of urban areas. According to the obtained findings in the present study, the local scale of the central neighborhood of the historical context has been found to be intelligible, while in the larger scale of the district, the irregular pattern of spatial configuration, especially in the main streets leading to the downtown area, have resulted in a mitigating intelligibility value. Thus, urban governance should pay particular attention to realizing the urban network system as a whole structure to enhance the level of legibility based on the relational characteristics of urban elements.

The results obtained from the subjective factors of legibility using cognitive sketch maps detected the significant association between the most important landmark and the position of highest value of integration in the syntactical analysis. This fact manifests the existing correlation between the objective and subjective facets of legibility. Hence, urban planners and designers should consolidate the most salient landmarks of the city with highly integrated locations of the city. This issue must particularly be taken into consideration in urban renovation and regeneration projects being carried out in historical districts. It should be noted that historical textures and monuments possess a substantial impact on the formation of cognitive maps in the minds of pedestrians. In fact, due to the antiquity and identity values of these historical monuments, as well as citizens’ sense of belonging to such urban elements, they are characterized by their undeniable role in the creation of mental images for pedestrians.

According to the obtained findings, there is a remarkable relationship between the objective and subjective aspects of legibility, and the formation of human mobility patterns. In fact, the incorporation of highly integrated configurational attributes, along with locating salient landmarks, can lead to the creation of crowded, sociable, and livable communities. This issue is specifically applicable in the areas where the creation of focal spaces is taken into account. The other important notion acquired from the empirical assessment of the study area was the citizens’ tendency towards participating in social activities at nighttime, and therefore that the most crowded human mobility patterns were observed between 8.15 pm and 9.15 pm. Accordingly, the authorities should draw particular attention to providing suitable services for pedestrians during these peak hours.

Additionally, the most observed constituted behaviors in these areas encompassed sitting, social interactions, and walking. Therefore, the quality of urban design in historical districts with the aforementioned characteristics encourages physical activities and the establishment of social interactions among the people visiting the target area, in addition to the stimulation of a livable urban environment. However, another notable notion affecting human mobility patterns in the study area was the constant physical presence of moral security agents (morality police) in front of the main landmark of the city in the time period between 6.00 pm and 10.00 pm. Empirical evidence indicated that this governing policy had a negative impact on the formation of human mobility patterns. Thus, some women tend to avoid using those spaces for this particular reason. As a result, these issues need to be taken into account in order to create socially sustainable urban areas.

### 4.3. Limitations and Orientation of Future Studies

One of the limitations for implementing the present research procedures was pertinent to hiring a sample size in relation to the community of experts for the acquisition of hand-drawn sketches. In fact, the process of identifying experts who were able to draw their mental maps appropriately and persuading them to dedicate at least 20 min of their time for accomplishing the task, along with them being profoundly familiar with the study area, was a challenging and time-consuming feat. Another restriction that had to be tackled in the present study was related to the execution of time-lapse photography methods. Due to the existing overcrowded population within the study area, and lack of feasibility for occupying the target spot during the entire period of conducting the empirical studies, it was impossible to use a tripod as a required instrument to ensure the highest possible quality of photos. Thus, the process of capturing photos was performed manually.

Furthermore, another constraint in this method was found to be the impossibility of conducting direct correlations between the obtained research data from each method. This is mainly due to the distinguished nature of the data. The present research design has focused on performing an empirical assessment towards facilitating the perception of spatial legibility, and its relationship with human mobility patterns, while the approach of future studies may rely on the association between spatial legibility and human behavioral patterns. Likewise, further nuanced research on the effect of visual barriers on social behaviors using the conformity of visibility graph simulations and empirical studies may enrich the existing literature. Moreover, other influential variables such as *sense of safety*, *wayfinding*, and *attraction of tourist hubs* can be measured through the present research design.

## 5. Conclusions

The primary objective pursued by developing this research was to provide an empirical assessment towards facilitating the perception of spatial legibility and its association with human mobility patterns. The attainment of legibility in existing urban areas has become a critical obstacle due to the widespread development of disintegrated actions and unconventional constructions. These drawbacks have led to incompatibilities in the urban network system. This study attempts to develop a novel mix method in order to further understand the association among syntactic, cognitive, and mobility patterns in urban areas as its contribution to the existing literature in this field. The results obtained from the present study indicated that the initial cores of historical urban districts in Rasht City, Iran possess a very poor spatial legibility.

In addition, by focusing on local historical districts, the range of legibility increases, whereas the analysis of the larger network configuration indicates a considerable mitigation in the value of legibility. These findings suggest that the landmarks featuring historical, vernacular, and socio-economic values are identified as the most important subjective factors affecting a legible urban environment, respectively. Moreover, the results revealed that there is a significant association between spatial legibility and human mobility patterns, so the incorporation of objective and subjective factors affecting legibility including highly integrated morphological characteristics, along with the saliency of landmarks featuring historical values, can lead to increased human mobility patterns. To recapitulate, adopting and enhancing spatial legibility in historical districts can create more lively and socially sustainable urban environments.

## Figures and Tables

**Figure 1 ijerph-19-15258-f001:**
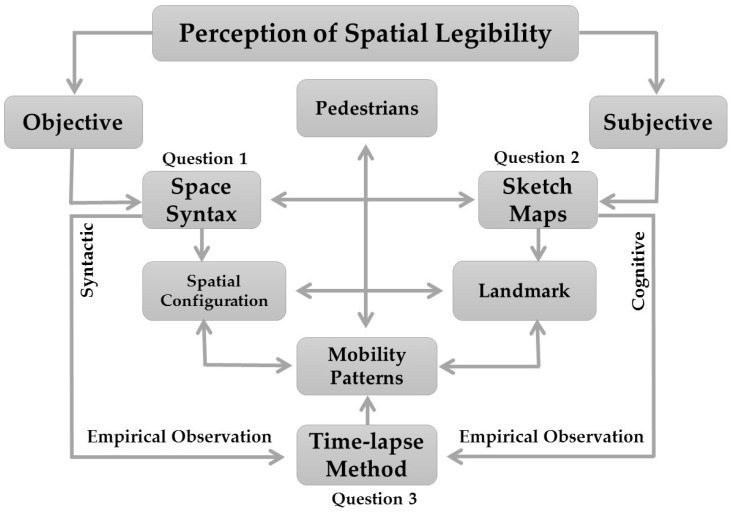
Research design and the functional interrelation of the adopted procedures.

**Figure 2 ijerph-19-15258-f002:**
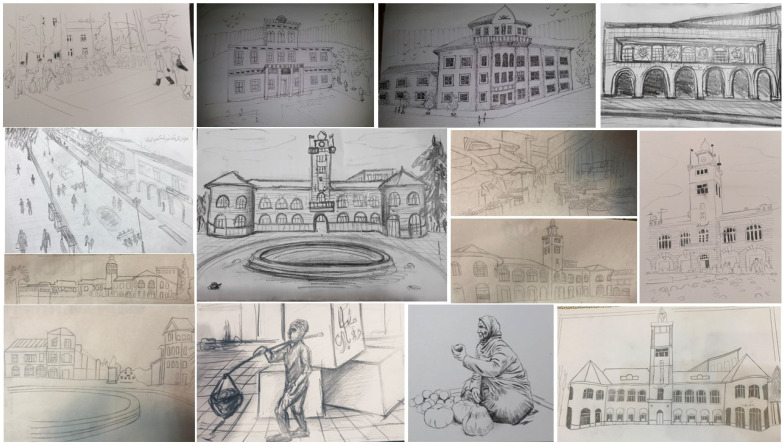
Some examples of the obtained hand-drawn sketches based on the acquired data from participants.

**Figure 3 ijerph-19-15258-f003:**
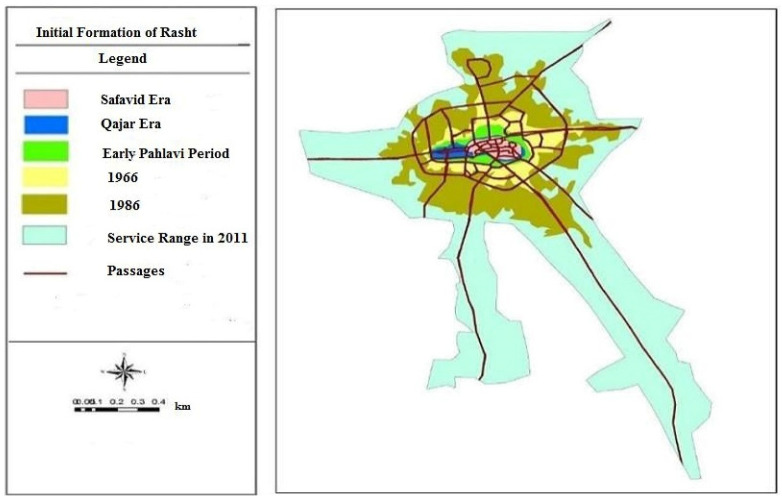
The initial pattern of urban development of Rasht in different dynasties [89].

**Figure 4 ijerph-19-15258-f004:**
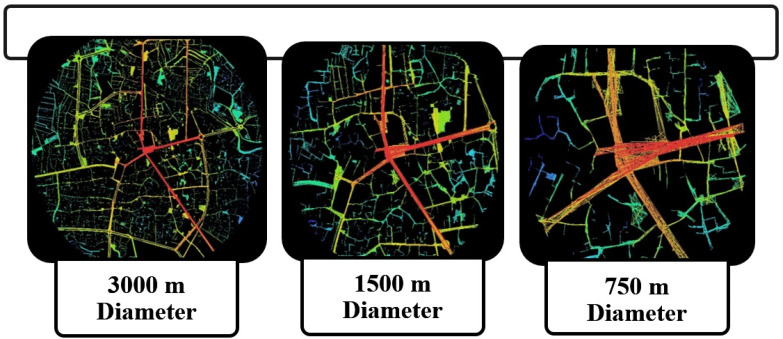
The spatial configuration analysis of the axial map graphs in different cases using the Space Syntax method, performed using the Depthmap software.

**Figure 5 ijerph-19-15258-f005:**
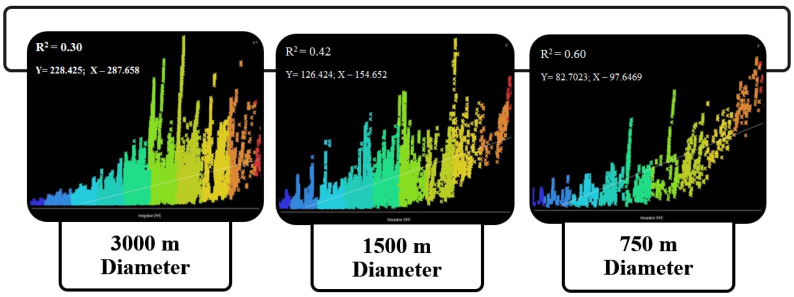
The spatial intelligibility values in different cases using the Space Syntax method, performed by the Depthmap software.

**Figure 6 ijerph-19-15258-f006:**
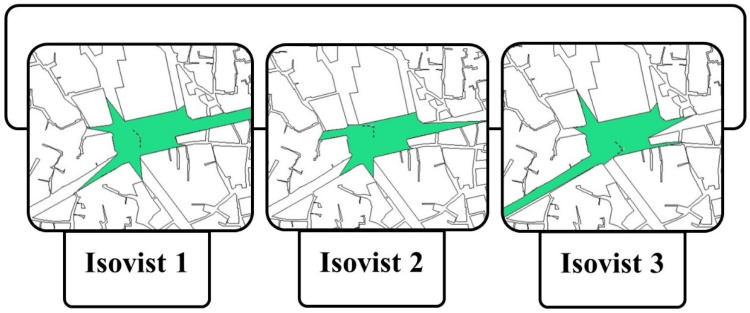
The obtained potential fields of view based on three positions towards the central square using Isovist graphs.

**Figure 7 ijerph-19-15258-f007:**
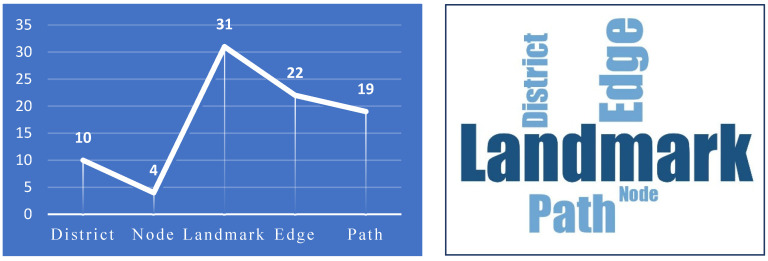
Frequency of identified codes, and the most significant elements influencing subjective legibility using the *word cloud processor* of the MAXQDA software.

**Figure 8 ijerph-19-15258-f008:**
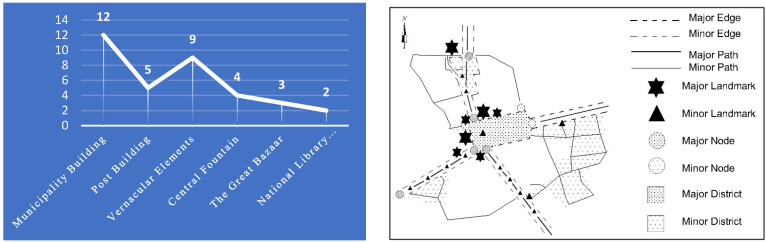
Identifying the most important landmarks based on the frequency of their usage in the hand-drawn sketches made by the participants, and the obtained cognitive map of the participants.

**Figure 9 ijerph-19-15258-f009:**
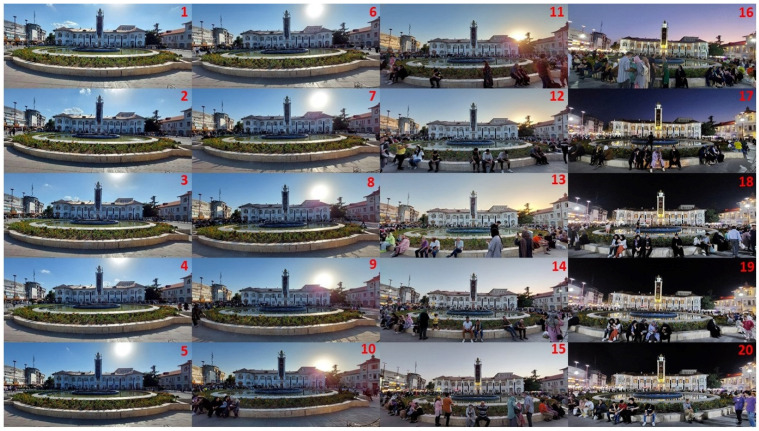
Pictures captured in front of the main landmark of the city between 5.00 p.m. and 10.00 p.m. using the time-lapse photography method. Each number was captured at a regular interval of every 15 min in order to follow up the possible transformations in the crowded communities.

**Figure 10 ijerph-19-15258-f010:**
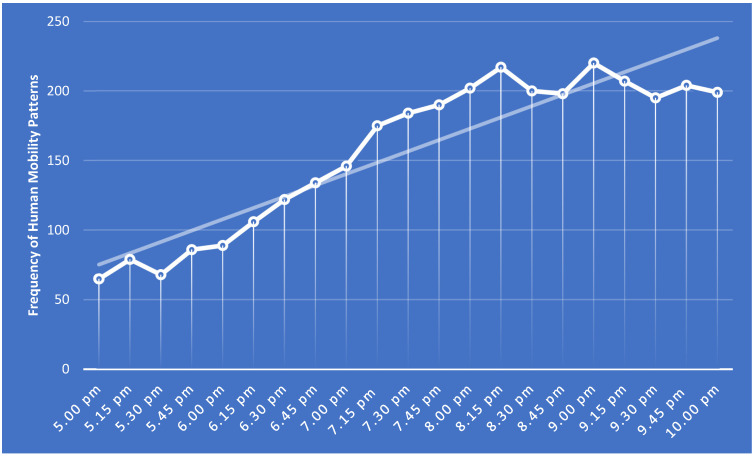
The correlation between different time periods and the frequency of human mobility patterns in the study area.

## Data Availability

Data is contained within the article.

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
