# Peer review of "Perception of Spatial Legibility and Its Association with Human Mobility Patterns: An Empirical Assessment of the Historical Districts in Rasht, Iran"

_ijerph, 2022, doi:10.3390/ijerph192215258_

Round 1

Reviewer 1 Report

The paper reports results from what the authors call a mix method study of legibility in a historical city, using the city of Rasht, Iran as a case-study. They find that (1) "significant correlation between spatial legibility and mobility patterns" and so argue that matching high integration areas with salient landmarks will produce "greater mobility patterns" (one assumes they mean 'more traffic', or there is no meaningful way to to interpret greater mobility patterns). Additional findings are that (2) "irregular configuration... mitigates level of legibility;" and that (3) "range of legibility" increases at local scale in comparison to the "value of legibility" at "larger network scale."

There are several problems with these findings, both in the way they are stated and with the substance of the claims themselves:

Finding (1) is clearly not accurate since they do not report any correlation between spatial legibility and mobility patterns. Their measure of spatial legibility only produces a single value for the entire city and what they call mobility pattern is not really mobility pattern, but simply change in average frequency of people observed over time (morning to evening) at a particular plaza or open space associated with a major landmark. 

Finding (2) does not seem to have any reference to their data or observations, and it is not clear what the claim is--why would irregular configuration mitigate (i.e. lessen the impact of) legibility? and why would that matter? Maybe the authors were not clearly able to express what they intended to say.

Finding (3) is derived entirely from three syntactical maps analysed at different scales. But there are several serious methodological problems in the procedures adopted by them. They do not specify how they selected the sub-system to analyze at each scale, or they they were able to account for the edge effects. there is also the problem of width of roads increasing with scale leading to greater number of axial lines in the central areas in the smaller scale map. While the finding seems intuitively right, the problems with procedures make the actual numbers obtained very unreliable. 

There are further methodological problems in the paper as well. What they claim are cognitive maps are not really cognitive maps, but rather aspects of the image of the city. In order to convert them into a 'map', several such images ought to be combined and mapped onto the layout of the city (as Kevin Lynch did). The procedure of having people draw what they recall is also suspect since it biases people to produce what they are able to draw. It is not a surprise that landmarks are reported more often because very few people would be able to draw districts or even nodes. So the data obtained in this procedure might simply be artifacts of the procedure rather than genuine capturing of the image of the city in the mind of the participants. The authors would be advised to take a second look at the number of procedural steps that Lynch took in order to ensure that the data he obtained from different techniques were consistent with each other, if they feel they need a model for their study.

Finally, the authors do not give any rationale or a framework for combining the different data collection methods they have used. The space syntax method produces one data point for the entire city (three if you read it at different scales); the recalled image is not mapped to the layout, and the count of people in front of the central monument collects data at a temporal scale. It is difficult to see on what basis would the three be combined.  

The paper does have a few strengths--the authors present a good review of the literature on spatial navigation and spatial cognition, and the organization of the paper is clear and transparent. Some good editing would clear away awkwardness of language that persists throughout. But the methodological problems are too serious for a recommendation to publish without a serious recasting of the entire paper.

Author Response

The authors would like to appreciate the reviewers for their valuable comments in order to improve the quality level of the manuscript. The revised manuscript has been systematically improved with new information and additional interpretations. It should be noted that revised items have been distinguished by red color in the manuscript text. Our responses to the referee’s comments are given below.

Reviewer 1

  1. Thanks for this comment! With respect to the first question, we guess that there is a misunderstanding in conveying the notion of the sentences written by the authors. As it is revealed in the title of the manuscript, this paper aims to provide an empirical assessment towards facilitating the perception of spatial legibility and its association with human mobility patterns. Thus, the written sentences in the Abstract and Conclusion, emphasizing on “correlation” have been revised and replaced with “association” in order to declare the relationship between these two items. For further clarifications, it should be noted that the measurement of spatial legibility has not only been conducted by a single value for the entire city. But also based on the conducted theoretical framework, in order to attain spatial legibility, two items of objective and subjective facets ought to be investigated. Accordingly, the combination of objective (using space syntax) and subjective (using sketch drawings) factors revealed that the target area of the plaza, in front of the main landmark of the city, is considered as the most legible place in this city. Therefore, the attainment of spatial legibility and human mobility patterns have been assessed through different methods and it is impossible to conduct a direct correlation between them. Hence, the process of accommodating the most legible place with the positioning of conducting time-lapse photography should be accomplished subjectively through inferential analysis. Of course this can be regarded as one of the limitations of the present research design; that is why, in respect to the comment of the respectful reviewer, this item has been added as one of the limitations of this study in the subsection 4.3 of the revised manuscript. It should be noted that, according to the given definition, human mobility can be defined as the ability to move and interact freely and easily among people which is able to be primarily assessed through empirical observations such as time-lapse photography. Thus measuring use frequency of the place makes sense here as which implies the association between legibility and social mobility. Additionally, this method offers stringent and reliable data in order to ensure validity of the obtained data over the time by considering the fourth dimension which indicates its superiority compared to other conventional methods. Moreover, in order to enrich the reliability of conducted methods, Isovist analysis has been conducted and added in the last paragraph of the subsection 3.1 in order to validate the adopted methods as well as possible.
  2. With regard to the second comment, the intention of the authors was not expressed properly in the former version. Therefore, this sentence has been revised and corrected in the Conclusion section. In addition, further descriptions and elaborations regarding the obtained findings from spatial configuration analysis were added to the second paragraph of the subsection 3.1 in order to clarify them as much as possible.
  3. Regarding the third comment of the respectful reviewer, the expression of implementing the research procedure has been revised in the last paragraph of the subsection 2.1. Admittedly, the mix-used methods adopted by this study are not that watertight, but more or less experimental. This is largely due to the fact that there are limitations of Space Syntax in current literature, mentioned by the Reviewer2 as well, thereby using multiple approaches, i.e., Space Syntax, cognitive sketch maps, and time-lapse photography, can complement these shortages to a large extent. Moreover, it should be noted that the method of implementing the Space Syntax in this research is based on the initial scheme of urban development in the city of Rasht which formed its historical configuration in different periods. Since the primary focus of this study is on historical districts of Rasht, attempts have been made to analyze the urban development in different historic periods. Accordingly, these three cases were analyzed to provide a tangible understanding in relation to the effect of spatial configuration on spatial legibility in different periods. Please bear in mind that there is no scale change within the analytical process of urban plans, but there is a change in extent diameter of the urban plans for the process of analysis. Therefore, there are no changes in width of roads and it is natural to have a greater number of axial lines in the larger analytical maps. In order to make above expressions more understandable for readers, all the vague statements in expression of procedures and results were revised in the revised paper as you pointed out (see P. 4).
  4. According to the comment of the respectful reviewer, the recalled hand-drawn sketches obtained from the participants were mapped into the urban layout in figure 8 in order to complete the process of cognitive mapping in the revised paper. Based on the multiple research resources, it seems not to be logical to address the participants about what they ought to be drawn (e. g. district and node), in order to avoid any sort of bias or orientations in their minds. Hence, they were initially asked to close their eyes and visualize the most prominent mental image that they recall in their minds in respect to the cityscape of the study area and then, convert their subjectivity to objectivity through drawing their cognitive mental images. Certainly, the authors considered the fact that different people have different drawing skills that may impact the outcome of the obtained data. That is why, the participants were limited to the community of experts in the field of architecture, urban design and graphic design. Prior to conducting interviews, the authors ensured regarding the participants' drawing skills and capabilities by observing their sample of works. It should be noted that further clarifications for the adopted research procedures were added in the subsection 2.2.
  5. Further elaborations regarding the rationale of combining the adopted procedures were added to the fourth paragraph of the subsection 4.1 of the manuscript.
  6. The linguistic quality of the manuscript has been double-checked and revised in terms of grammatical issues by a proficient native editor for all parts of the manuscript in order to ensure the fluency and suitability of the manuscript. Thank you again for your extensive comments which help us greatly to improve the paper.

Reviewer 2 Report

Summary

The limitations of Space Syntax due to the necessary simplification (or, i.a., the impact of the edge effect on the test results) are well-known and recognized. Hence the search for extensions of methods and tools to get a more complete and meaningful picture. This study attempts to develop a novel mixed methods approach to understanding the relationship between syntactic, cognitive, and mobility patterns in urban areas as its contribution to the existing source literature.

General comments

The authors identify the research gap, and their study aims to fill it and link it to the existing literature on the subject. In this regard, time-lapse photography, used as one of the techniques, is a thought-provoking approach to the problem of modeling the behavior of users of public spaces. The authors rightly note that the fourth dimension- time- is also essential in addition to the three dimensions of the city. The reader's attention is drawn to the extensive appendix bibliography of nearly a hundred items. On the one hand, finding fresh thought in a rather intensively researched issue is challenging.

Specific comments 

Point 1. 

A minor note on Figure 1: some of the graph lines are skewed (e.g., under the ‚Empirical Observation' heading at Question 3) which disturbs a bit.

Point 2. [112-135] 

The analysis of the axial map is primarily concerned with the flow of people and their movement within the configuration of the urban network (or, in other words, the urban matrix). In addition to the Depthmap graph analysis, wouldn't it be a good idea to point out also the Isovist approach in these ‚convex spaces' where people meet?

For the study, it would be good to mention Isovist to analyze the legibility of plaza configurations and ease of orientation in public spaces. With such a simulation, it is possible to determine the theoretical level of balance between what is seen at a given location and what could potentially be seen through traffic from that location. The Co-Visibility (Rv) indicates the degree to which a person in a given place is involved in the interchange of "seeing and being seen." It seems to be beneficial in describing social behavior or lack thereof. It might be as well noted in the preliminaries of the follow-up research (‚Limitations and orientation of future studies.')

Point 3. [221-245] 

In part 3.1. 'Findings obtained from spatial configuration analysis,' I would suggest elaborating a bit more on the obtained values and indicators to more clearly discuss their names, enigmatic in nature, with what stands behind them about the reality and morphology and condition of public spaces in Rasht.

Point 4. 

Note on Figure 8: shouldn't it be considered to blur the images of people's faces to ensure anonymity?

Point 5. [285-307] 

The mere presence of people accounts for one of the essential factors of urban life, called livability. Could the authors add some two sentences characterizing what social interactions take place in the spaces of described cases? It would be helpful to link more closely to the Special Issue theme ("The Impact of Urban Design on Physical Activity and Social Interaction").

Author Response

The authors would like to appreciate the reviewers for their valuable comments in order to improve the quality level of the manuscript. The revised manuscript has been systematically improved with new information and additional interpretations. It should be noted that revised items have been distinguished by red color in the manuscript text. Our responses to the referee’s comments are given below.

Reviewer 2

  1. Figure 1 has been revised according to the viewpoint of the respectful reviewer.
  2. The authors are grateful for the intelligent comment of the reviewer in respect to enrichment of analysis using Isovist graphs. Accordingly, Isovist graphs were carried out and performed in order to reinforce the analysis as well as possible. Additional descriptions and interpretations of obtained results were added in the third paragraph of the subsection 2.1, and the last paragraph of the subsection 3.1. In addition, further insights regarding the effect of visibility graphs on social behaviors for the orientation of future studies were added in the subsection 4.3.
  3. Further elaborated descriptions regarding the findings obtained from syntactical analysis were added to the second paragraph of the subsection 3.1.
  4. All the recognizable faces have been blurred in the figure 8 for the anonymities of the people.
  5. Further descriptions regarding the established behaviors in the target area as well as the influence of urban design on physical activities and social interactions were added to the subsection 4.2.